# A Multi-Sensory Guidance System for the Visually Impaired Using YOLO and ORB-SLAM †

Zaipeng Xie [1,2,*,‡] , Zhaobin Li [2,‡], Yida Zhang [2], Jianan Zhang [2], Fangming Liu [2] and Wei Chen [2]

[1] Key Laboratory of Water Big Data Technology of Ministry of Water Resources, Nanjing 211100, China
[2] Department of Computer Science and Technology, Hohai University, Nanjing 211100, China; lizhaobin@hhu.edu.cn (Z.L.); zhangyida@hhu.edu.cn (Y.Z.); jianan_zhang@hhu.edu.cn (J.Z.); fangming_liu@hhu.edu.cn (F.L.); hachi10086@hhu.edu.cn (W.C.)
[*] Correspondence: zaipengxie@hhu.edu.cn
[†] This paper is an extended version of our paper published in the 8th International Conference on Progress in Informatics and Computing (PIC-2021), Shanghai, China/Tampere, Finland, 17–19 December 2021.
[‡] These authors contributed equally to this work.

**Abstract:** Guidance systems for visually impaired persons have become a popular topic in recent years. Existing guidance systems on the market typically utilize auxiliary tools and methods such as GPS, UWB, or a simple white cane that exploits the user's single tactile or auditory sense. These guidance methodologies can be inadequate in a complex indoor environment. This paper proposes a multi-sensory guidance system for the visually impaired that can provide tactile and auditory advice using ORB-SLAM and YOLO techniques. Based on an RGB-D camera, the local obstacle avoidance system is realized at the tactile level through point cloud filtering that can inform the user via a vibrating motor. Our proposed method can generate a dense navigation map to implement global obstacle avoidance and path planning for the user through the coordinate transformation. Real-time target detection and a voice-prompt system based on YOLO are also incorporated at the auditory level. We implemented the proposed system as a smart cane. Experiments are performed using four different test scenarios. Experimental results demonstrate that the impediments in the walking path can be reliably located and classified in real-time. Our proposed system can function as a capable auxiliary to help visually impaired people navigate securely by integrating YOLO with ORB-SLAM.

**Keywords:** multi-sensory guidance system; visually impaired; ORB-SLAM; YOLO; point map building; indoor navigation

## 1. Introduction

Research on guidance systems for visually impaired people has become a popular topic in recent years [1–5]. Vision impairment may restrict one's mobility due to its reduced capability to detect obstacles [6]. Nevertheless, individuals with severe visual impairments may travel independently, assisted by a wide range of tools and techniques [3]. The ability to navigate appropriately can be greatly assisted by familiarity with an environment or route. However, indoor navigation in unknown environments is still inefficient due to the diversity of indoor environments. Obstacles in indoor environments can be small and not easily explored by the visually impaired person, and indoor spaces can be unpredictable. Therefore, further research is desired to explore alternative techniques that enable visually impaired persons to function effectively and efficiently. Conventional methods to facilitate a visually impaired person involves tools such as a long cane or a guide dog. A long cane can increase the user's range of touch feeling and is frequently swung in a sweeping downward motion across the intended direction of passage. However, cane travel experiences can differ depending on the user and the situation. Guide dogs can be limited by their inability to understand complex directions. Extensive training [7] is usually required for guide dogs,

and their service life may not exceed a few years. Hence, both techniques may not be an ideal solution for the blind community.

The usage of GPS devices [8,9] as mobility assistance has become popular for outdoor navigation tasks. However, research demonstrated that it is not an adequate substitute for traditional mobility aids such as white canes and guide dogs. This is because, although most GPS trackers can provide latitude and longitude on a minute-by-minute level, GPS can be ineffective in a complex environment due to low signal strength and low accuracy. Ultrasonic positioning sensors [10] can be used as an additional facility to sense the surroundings and detect upcoming potholes. Nevertheless, the performance of such systems can be volatile due to the variation in real-time conditions. Ultra-wideband (UWB) indoor positioning [11] relies on transmitting extremely short pulses to measure the time of arrival (ToA), angle of arrival (AoA), and received signal strength (RSS). These signals can be used to reconstruct the positioning of a person wearing a UWB device. However, they require extensively deployed positioning anchor points that may be costly and difficult to maintain.

Simultaneous localization and mapping (SLAM) [12] based guidance systems can be a viable solution for visually impaired persons. SLAM-based systems can determine the position and orientation of the sensor relative to the surrounding environment and reconstruct the mapping simultaneously. Visual SLAM [13] is a particular type of SLAM system that can implement the function of SLAM positioning and mapping using 3D cameras. Existing research on the SLAM-based blind navigation technology generally combines visual SLAM with other optimization methods. Kanwal et al. [14] use a Kinect somatosensory system to combine the RGB-D camera with the image information obtained by infrared sensors. Bai et al. [15] integrate a virtual blind road following method based on oriented fast and rotated brief SLAM (ORB-SLAM) [16] to provide automatic navigation for visually impaired people. However, their method relies on optical perspective glasses to provide feedback on the navigation results through vision and hearing. Chen et al. [17] combine the semantic segmentation method to obtain semantic information in the environment and provide voice feedback to the blind. However, these methods mentioned above mainly utilize a single sensory function for navigation guidance, and the effectiveness can be limited in a noisy environment. A system simultaneously combining the tactile and auditory senses with an active SLAM can be more intuitive in facilitating the mobility of visually impaired users.

This work proposes a multi-sensory indoor navigation guidance system by combining ORB-SLAM, YOLO-based target detection [18,19], an RGB-D camera, an audio speaker, and mechanic motors. Unlike the existing visual SLAM navigation system, In our proposed system, real-time local navigation is provided by detecting the surrounding obstacles through the RGB-D camera. Meanwhile, the global navigation function is implemented by synchronously positioning the ORB-SLAM on the reconstructed map. A prompt vibration motor conveys obstacle directions to the visually impaired user. In addition, our proposed system can recognize the obstacle using the YOLO-based target detection. The obstacle information in the global navigation direction can be prompted through a mini-speaker. The combination of the three proposed functions brings forth the multi-sensory guidance system for the visually impaired.

The main contributions of this paper are as follows:

- We evaluate the potential of ORB-SLAM and YOLO as assistive methods for the blind guidance system. A multi-sensory blind guidance system is proposed and implemented to combine both the tactile and auditory sensations;
- We propose improving the conventional SLAM accuracy by generating a dense navigation map. Our proposed system utilizes the position information from the ORB-SLAM that can be fused using the Bresenham algorithm [20]. A dense navigation map is developed by transforming coordinates to feature points. A local obstacle avoidance algorithm is developed to identify short-range obstacles through point cloud filtering and angle sampling;
- We implemented the overall system as a prototype for our proposed equipment. Experiments are performed in four different environments. Results demonstrate

that our proposed system can accurately reconstruct the map of the surrounding environment in various scenarios. Our system is lighted and weighted and does not require an external power supply. Trial experiments show that it can be useful auxiliary equipment for the community of vision impairment and enables the visually impaired person to move about safely.

## 2. Related Work

Researchers have worked on various prototypes of guidance systems for visually impaired people. Santiago et al. [21] surveyed the development of systems using traditional technologies such as sonar, infrared radiation, and inertial sensing. The authors conclude that smartphones and wearable devices can be valuable as assistive devices for individuals with visual impairments. Plikynas et al. [3] surveyed the utilization of industrial indoor navigation technologies such as near-field communication (NFC) and radio-frequency identification (RFID) as indoor navigation tools for the visually impaired. Walter et al. [22] reviewed various indoor positioning-based systems, including WiFi positioning, inertial Sensors, and light and sound-based systems, for navigation of the visually impaired.

Artificial intelligence technology has emerged as a good candidate for assisting indoor navigation technology in recent years. Budrionis et al. [23] surveyed smartphone-based computer vision traveling aids, and the authors argue that assistive devices for people with visual impairment should be affordable for general users. Huang et al. [24] proposed a vibrotactile feedback-based indoor navigation robot to assist navigation using haptic information. The robotic system can effectively provide an anti-interference capability. However, the hardware cost is high and may not fully solve the user experience optimization problem. Zhao et al. [25] propose a smart glasses navigation system for low-vision people to provide visual and audio pathfinding guidance. This system works well for low-vision people, but the results for sightless people need to be improved. In addition, the system can only provide limited battery time due to the size constraint. Romeo et al. [26] propose an intelligent assistive system that performs environment exploration and path planning through a force feedback tablet and transmits information through a haptic belt. The system is acceptable for navigation and obstacle avoidance, but it is not convenient enough and requires several components that may not function synchronously.

Recognizing obstacles through sound is another major research direction for assisting visually impaired people. Metsiritrakul et al. [27] propose a stereo detection system. This system can analyze the non-verbal stereo information to acquire details such as direction, distance, and size for obstacle identification. Chih et al. [28] propose an indoor localization system for blind people based on ultrasonic sensing technology. It can identify and detect obstacles with low complexity and reduced power consumption. Visually impaired people can avoid obstacles using such a system via auditory and tactile information. Presti et al. [29] proposed an ultrasonic obstacle detection system that can estimate the distance and location of obstacles. These systems rely on dedicated hardware and may not be able to recognize head-level obstacles.

Another interesting research direction on assisting visually impaired people is redesigning the traditional aid of a blind cane to improve the lives of people with visual impairment. Chang et al. [30] proposed an intelligent assistive system based on wearable smart glasses and a smart cane that can detect aerial obstacles and fall events on the road. Their smart cane can guide the user by vibrating to avoid aerial obstacles and collision accidents and promptly notify the user's family when he or she falls. Masud et al. [31] propose a smart cane consisting of a Raspberry-Pi 4B, an RGB-D Camera, and an ultrasonic Sensor. Their smart cane can recognize obstacles by simple calculations and provides hints by vibration. This system's object recognition and obstacle avoidance effect are acceptable. However, it does not provide real-time navigation, and it lacks memory of visited locations and hence may not be able to provide the users meaningful advice in familiar areas.

In summary, current assistive devices for visually impaired people have limited functions. Most researchers focus on obstacle avoidance, with relatively little research on indoor

path navigation and object recognition. Moreover, some research results have problems such as high power consumption and low user acceptance, which cannot fully meet user needs. Considering the recent development experience of devices for visually impaired people, we propose an intelligent cane system based on the YOLO algorithm and the ORB-SLAM system, which integrates and improves obstacle avoidance functions, indoor navigation with memory, and object recognition. This system communicates with users by providing auditory and tactile information. Hence it can satisfy the major requirement of visually impaired people and act as a competent auxiliary to enable users to move about safely.

## 3. System Principles

Figure 1 describes the algorithm flow of the proposed multi-sensory guidance system. Our system consists of two inputs. They are the RGB-D video frames acquired by the depth camera and the user's audio input. First, all inputs are sent to the controller process that can turn on and off other processes based on the audio input. The video frames are sent to the SLAM mapping, the local navigation, and the object detector processes. The navigation process calculates the optimized path based on the dense map and the forward direction information is also produced. The direction information is fed to the fusion process. The local navigation process builds a local map based on the input video frames, calculates the obstacle-free direction and then transmits the direction information to the fusion process. The fusion process generates the optimized forward direction by fusing the directional information and finally turns it into vibration signals.

In addition, the video frames are sent to the object detector process to provide information about the objects in the scene via the YOLO-based target detection model. Audio hints can be produced to facilitate the user in making decisions.

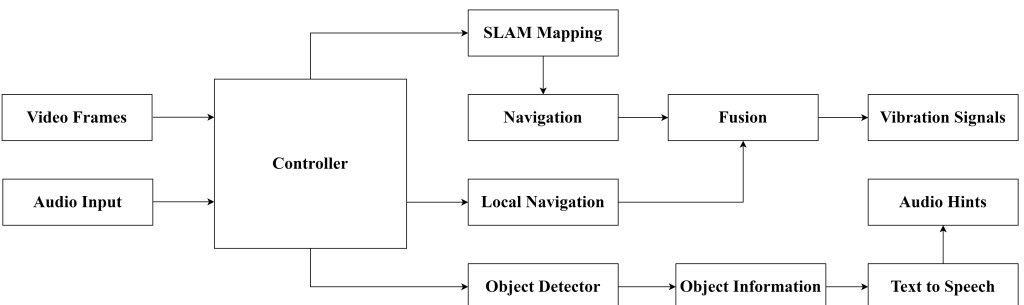

**Figure 1.** Algorithm flow of the proposed multi-sensory guidance system.

### 3.1. SLAM Mapping

The SLAM mapping process is to reconstruct a dense point cloud map for navigation. We utilized the ORB-SLAM framework [16] to obtain the sparse point cloud map of the user's environment. Figure 2 shows the structure of the ORB-SLAM framework, and it has three main components: tracking, local mapping, and loop closing. Tracking is responsible for the localization and monitoring of each frame acquired by the RGB-D camera, which is used for pose estimation and optimization by matching feature points. Local mapping is responsible for inserting and deleting keyframes and feature points. It also provides optimization for local keyframes and map points. The loop closing section searches for each detected keyframe, and once the presence of the loopback is identified, the loopback is aligned and fused using a similarity transformation.

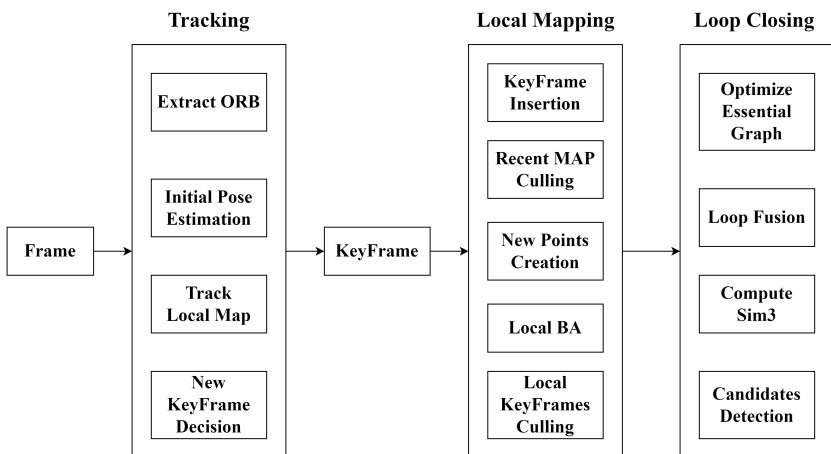

**Figure 2.** A block diagram of the data flow in the ORB-SLAM process.

However, the point cloud maps produced by a conventional ORB-SLAM are sparse and may not be used directly for path planning and navigation. We deploy the robotic operating system (ROS) nodes for creating dense point cloud maps. For each keyframe in the building process, we iterate through the depth matrix of the keyframe. Each depth point in the matrix represents its position in the camera coordinate system. We need to transform it into the coordinates in the world coordinate system [32]. The transformation formula is as follows.

$$
\begin{cases}
z = d \\
x = \frac{i - C_x}{f_x} z \\
x = \frac{j - C_y}{f_y} z,
\end{cases}
\tag{1}
$$

where $C_x, C_y, f_x, f_y$ are intrinsic parameters of the RGB-D camera. Since the world coordinate system is right-handed, we add depth points with depth $z$ within the range $[0.05, 6]$ meters and height $y$ within the range $[-0.6, 0.3]$ meters to the dense point cloud map. After all keyframes are calculated, the dense point cloud map can be produced.

Assuming that some discrete error points exist in the dense point cloud map, we use a statistical filter to process the point cloud map in order to improve navigation efficiency. The general idea can be summarized in two steps:

1. For each point in the map, we calculate its average distance $d$ to the nearest $K$ points. This calculation is iterated for each point in the input point cloud. Therefore, an array containing the $d$ values of each point can be obtained, denoted as distance;

2. For all the points in the input point cloud, it is assumed that the elements in the distance array follow a Gaussian distribution, the array is a sample, and the sample capacity is the number of points in the point cloud. The shape of the Gaussian distribution curve is determined by the mean and standard deviation of the sample, and the corresponding points whose $d$ values are outside the standard range are considered outliers and will be removed.

After obtaining the dense point cloud map, we rotate and project it to the $(X, Y)$ plane to generate the occupancy grid map employed for navigation.

### 3.2. Navigation and Path Planning

Right before the system lays out the navigating plan for the user, we need to obtain an optimized path from its current location to the target location. Several methods [33], such as Dijkstra, BFS, and DFS algorithms, are used for planning paths in planar grid maps. We develop the path planning algorithm (PPA) based on the A-Star algorithm. Compared with some popular algorithms, our proposed PPA algorithm is based on the greedy scheme that can find the shortest path quickly. Our PPA algorithm can not only consider the distance

from the node to the starting point but also the distance from the node to the endpoint as the cost function. The pseudocode of the proposed PPA algorithm is described in Algorithm 1. In the PPA algorithm, assuming the number of nodes in the graph is $n$, the outer loop of the algorithm obtains the point with the lowest score from the graph each time, and its time complexity is $O(n \cdot \log n)$ due to the need for sorting. The inner loop of the algorithm iterates through all the neighbors of the current node, calculates its score and updates it. The overall time complexity of the algorithm is $O(n^2 \cdot \log n)$. In addition, the algorithm needs to store all node information in the graph, and its space complexity is $O(n^2)$.

After the optimal path is established, the system is designed to provide directional advice. We develop an orientation correcting algorithm (OCA) based on the Bresenham algorithm. The OCA algorithm can draw a circle with the user's current position as the center and an adjustable radius (typically 2 m) in the grid map to determine whether the user is currently in the vicinity of the path. If there is an intersection between the circle and the path, the vector formed by the user's current position and the intersection position is the directional guidance corresponding to the orientation, and we use the angle between the user's current and desired orientation as the navigation result to give instructions; if there is no intersection, the user is deemed to deviate from the correct path, and an optimal path is recalculated.

---

**Algorithm 1:** The proposed PPA Algorithm.

1 **Input:** graph, startNode, endNode
2 **Output:** path
3 **for** *node in graph* **do**
4     node.score := Inf;
5     node.heuristicScore := Inf;
6     node.visited := false;
7 **end**
8 startNode.score := 0;
9 startNode.heuristicScore := 0;
10 **while** *true* **do**
11     currentNode := nodeWithLowestScore(graph);
12     currentNode.visited := true;
13     **for** *nextNode in currentNode.neighbours* **do**
14         **if** *nextNode.visited == false* **then**
15             new.Score := calculate(currentScore, nextNode);
16             **if** *newScore ≤ nextNode.score* **then**
17                 nextNode.score := newScore;
18                 nextNode.heuristicScore := newScore + calculate(nextNode, endNode);
19                 nextNode.routeToNode := currentNode;
20             **end**
21         **end**
22     **end**
23     **if** *currentNode == endNode* **then**
24         return buildPath(endNode);
25     **end**
26 **end**

---

As shown in Figure 3, the black arrow indicates the current orientation of the user, and the red arrow indicates its orientation after planning. The left side depicts that there is no intersection between the circle and the path, and the path needs to be planned again; the right side shows that there is at least one intersection between the circle and the path, and the line between the center of the circle and the nearest intersection from the endpoint can be chosen as the planned orientation.

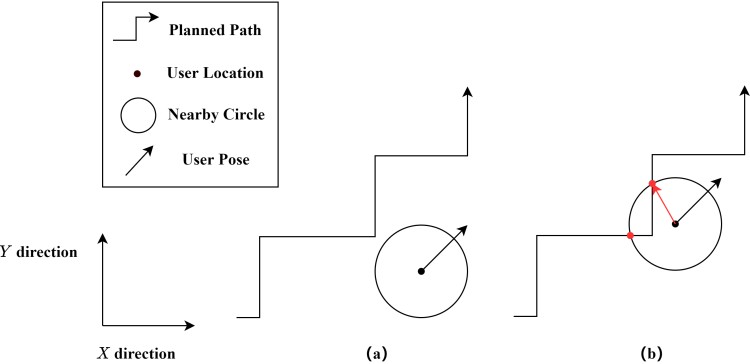

**Figure 3.** An illustration of the Orientation Correcting Algorithm. In situation (**a**), there is no intersection between the circle around the user and the planned path, thus an optimal path is recalculated. In situation (**b**), there is at least one intersection, and then we choose the closest intersection to the planned path's destination as the current goal and provide the user with the corresponding orientation.

### 3.3. YOLO-Based Object Detector

The YOLO target detection algorithm [18] has a simple structure, high computational efficiency, can facilitate end-to-end training, and has a wide range of applications in the field of target detection. In this work, the YOLO algorithm is used to detect the obstacles encountered by the visually impaired person while traveling and transmit the information to the user through the free exploration function and text-to-speech conversion function to achieve the purpose of obstacle avoidance. The YOLOv3 network structure consists of three parts: the feature extraction network, the feature pyramid, and the prediction network. The feature extraction network can improve the depth of the web with the help of the DarkNet53 network and provide a fast and accurate object detection solution for our system.

### 3.4. System Composition

At the hardware level, the system consists of an Intel RealSense RGB-D camera [34], a Raspberry Pi embedded board, a battery, and a vibration motor array. We use the robot operating system (ROS) [35] to integrate each functional module and manage the data transmission at the same time. The system consists of eleven ROS nodes at the software level, and they communicate mainly through the Publish–Subscribe Mode. From the I/O's perspective, the nodes of this system can be divided into three categories: input nodes, processing nodes, and output nodes.

The input nodes include the depth camera node and the speech recognition node. The depth camera node acquires and publishes the current color image frame and depth image frame in real-time; the speech recognition node acquires the user's voice input and converts it into text information.

The processing nodes are divided into four parts: control, recognition, map building, and navigation. The control node controls the opening and closing states of all processing nodes, and different state combinations correspond to different system modes. The recognition node acquires color image frames in real-time and obtains item information in images through target recognition. The map building node accepts depth image frames in real-time, creates a dense point cloud map by ORB-SLAM, and projects it to generate a planar occupation grid map. The navigation node includes the global navigation node, the fusion node and the local navigation node. The local navigation node builds the local point cloud map by depth image frame and releases the direction information without obstacle; the global navigation node calculates the best path between the user's current location and the destination and releases the prompt direction; the fusion node combines the information from both the local and global navigation nodes. The global navigation

node is responsible for establishing the best path between the user's current location and the destination and releases the directional advice.

The output nodes include a vibration node and a speech synthesis node. The output nodes can generate vibration indicating the direction without obstacles providing a tactile reminder for the user. The speech synthesis node can prompt the information of the received texts by audio to facilitate the user with auditory cues.

Overall, we build the system based on ROS with a modified ORB-SLAM method and the results are projected into a planar map. Then we utilize the proposed PPA algorithm to construct the shortest path and provide navigation advice. In addition, we employ the YOLO target detection algorithm to detect surrounding objects at the scene and guide the user from both tactile and auditory aspects. Hence, we integrate a multi-sensory guidance system for visually impaired users.

## 4. Experimental Results

### 4.1. Experimental Setups

The system is a multi-sensory assisted guide system based on edge intelligence. It is built using a ROS platform, with eleven nodes working together to achieve map building, navigation, target detection, and human-computer interaction, including audio and tactile information exchanging. For the hardware part, the test system consists of a development board Raspberry Pi capable of running a standard Linux system, an RGB-D camera, a vibrator composed of multiple groups of motors, and power supply facilities. The Raspberry Pi is equipped with a Linux system to run each ROS node in the system. The RGB-D camera uses a visual SLAM algorithm to receive color images and depth information to obtain the current color image frame and depth image frame in real-time. The experimental setup parameters are listed in Table 1.

**Table 1.** The Experimental Setup Parameters.

| Type | Parameter |
| :---: | :---: |
| Camera | Intel RealSense D435 [34] |
| Hardware Platform | Raspberry Pi 4B |
| Operating System | Ubuntu 20.04 LTS |
| Software Platform | ROS2 Galactic Geochelone [35] |

In Figure 4, we install the Raspberry Pi in the middle of the cane handle, and the power supply is placed under the board of the Raspberry Pi. The motor set is installed in the grip of the cane to provide tactile navigation messages for the user. The camera is installed in the upper part of cane handle at the right height to collect a complete picture of the environment. The camera is mounted on the upper part of the cane handle, the size of which can collect environmental information thoroughly. The primary motivation for the structural framework of this cane is to design a portable and functional smart device with upgradability.

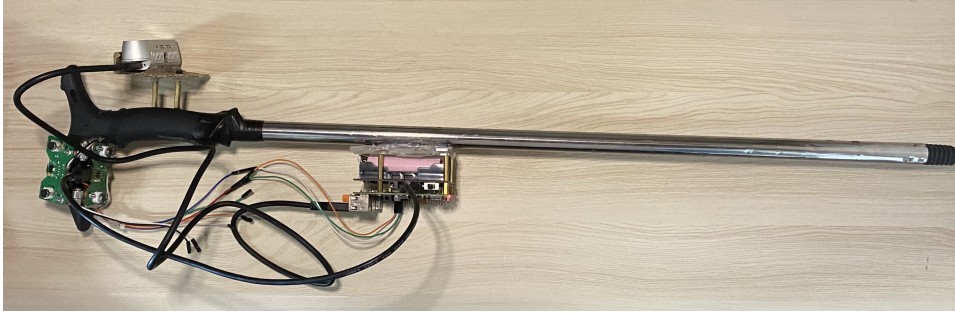

**Figure 4.** The physical diagram of proposed hardware system.

### 4.2. Map Building Experiments

#### 4.2.1. Experimental Scene Selection

Four scenes are used for the experiments as shown in Figure 5, from left to right, namely (A) laundry room, (B) H-shaped hallway, (C) classroom, and (D) T-shaped hallway. This label refers to each subsequent scene. The black line indicates the path of the experimenter during the experiment. Each of the four scenes has different characteristics, which can better simulate the indoor scenes that the visually impaired users may encounter and make the experimental data representative.

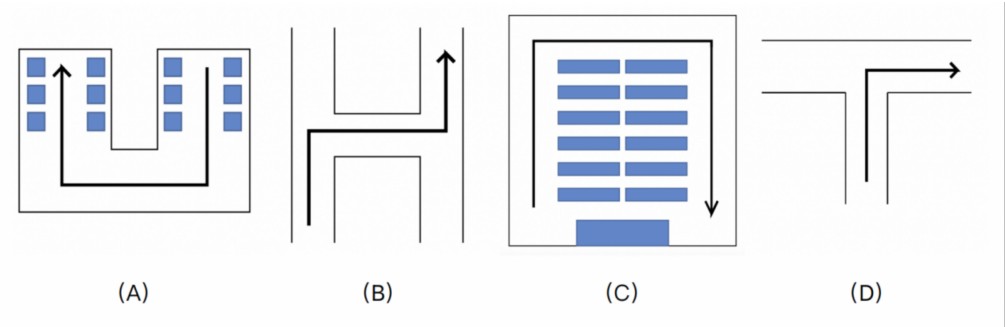

**Figure 5.** The experimental scenes. (**A**) U-shaped hallway; (**B**) H-shaped hallway; (**C**) Loop-shaped hallway; and (**D**) T-shaped hallway.

#### 4.2.2. Field Simulation

A user (Figure 6) is designated by wearing an eye mask to simulate a visually impaired user to test the functions of scene reproduction, free exploration, and global navigation. The realistic reflection of the various situations that the visually impaired users may encounter enhances the realism of the experiments.

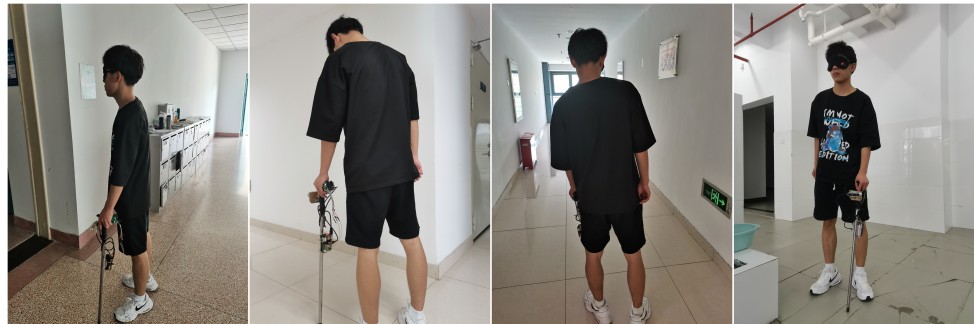

**Figure 6.** Example diagram of field experiment.

#### 4.2.3. Point Cloud Map Establishing Experiments

When a user walks with our proposed smart cane for the first time, the Mapping node of the system invokes ORB-SLAM real-time positioning and extracts identifiable feature points in the real-time RGB image of the Camera node. Then it pulls the corresponding depth information in the real-time depth image of the Camera node to achieve obstacle recognition, user location, and orientation positioning, and it draws a real-time 3D point cloud map in space based on the above information. In the experiments for four different scenes, the process of feature point labeling and point cloud map building is shown in Figures 7 and 8.

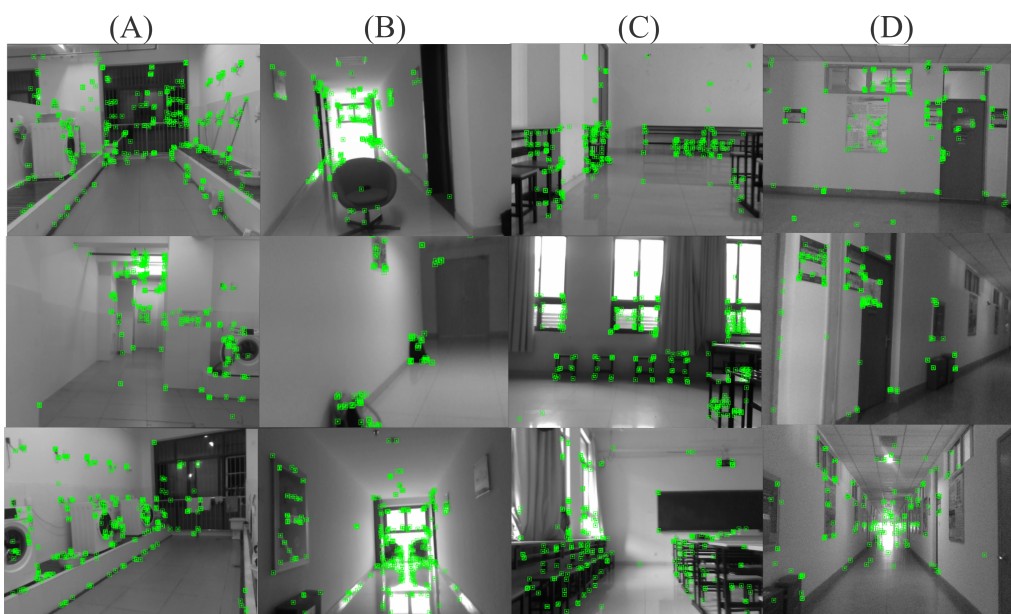

**Figure 7.** Example of the feature points extraction for (**A**) the washing machine and other objects in U-shaped hallway scene; (**B**) the obstacles in H-shaped hallway scene; (**C**) the seats and desks in the Loop-shaped hallway scene; (**D**) the walls in T-shaped hallway scene.

Figure 7 shows that the system identifies feature points in everyday indoor scenes in a stable and continuous process. It can locate ground obstacles in the settings and wall objects and identifiable attributes in a real-time and accurate manner to identify the barriers. It can capture the identifiable feature points in the image and locate the user's position and orientation in real-time with the change of depth information of the feature points to ensure real-time monitoring of the user's position in the scene. The 3D point cloud generating process built by ORB-SLAM is shown in Figure 8.

During the user's walk, the Mapping node captures the user's position and orientation in real-time through feature point recognition and operation and records them in the point cloud map. When establishing the point cloud map, the system indicates the user position and orientation with a blue square status and direction, a green square indicates the current situation and exposure, and a red square indicates the position and orientation when starting the system. The Mapping node can record the location and depth information of feature points in the 3D point cloud map in real-time. It also filters out the interference feature points the depth of which is inappropriate.

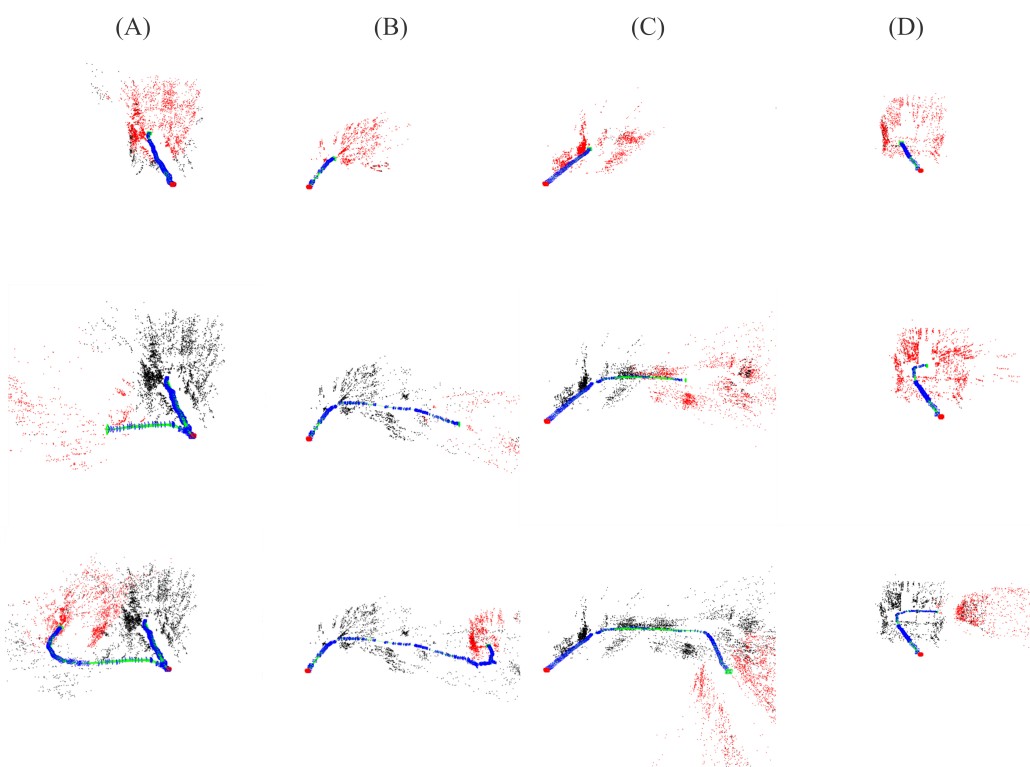

**Figure 8.** Example of the 3D point cloud of (**A**) the U-shaped hallway; (**B**) the H-shaped hallway; (**C**) the Loop-shaped hallway; (**D**) the T-shaped hallway.

The Local_Navigation node divides the depth camera field of view into 18 directions, reads the depth and position of the feature points, and counts the number of feature points falling in each direction. The direction is considered unavailable when the number of feature points exceeds a threshold value. In contrast, the rest of the area is considered available. After that, the Fusion node produces the conceivable direction according to the local and the global navigation data. The Vibrator node gives the vibration prompt of the corresponding direction according to the available direction to facilitate the user's perception of the environment.

4.2.4. Dense Point Cloud Map and Occupied Grid Map Reconstructing Experiments

During the local navigation process, the Mapping node further processes the point cloud map by filtering out the feature points that are out of range. Then it organizes the remaining feature points to form a dense point cloud map that can accurately reflect the obstacle. An example of the dense point cloud map is shown in Figure 9.

From the comparison between the four experimental scenes and the hand-drawn scene obstacle map, the dense point cloud map for the indoor settings can accurately record the scene's obstacle location, formulating a complete scene map. After that, the Navigation node receives the dense point cloud map of the scene and uses the auxiliary node OutoMap_Server to generate the occupied grid map. The final populated grid map generated by the experimental scenario is shown in Figure 10.

The final occupancy grid map is formed with black areas for obstacles and gray regions for ground, i.e., walking areas. This significantly reduces the occupied storage space while preserving the obstacle information so that it can be used later for subsequent global navigation.

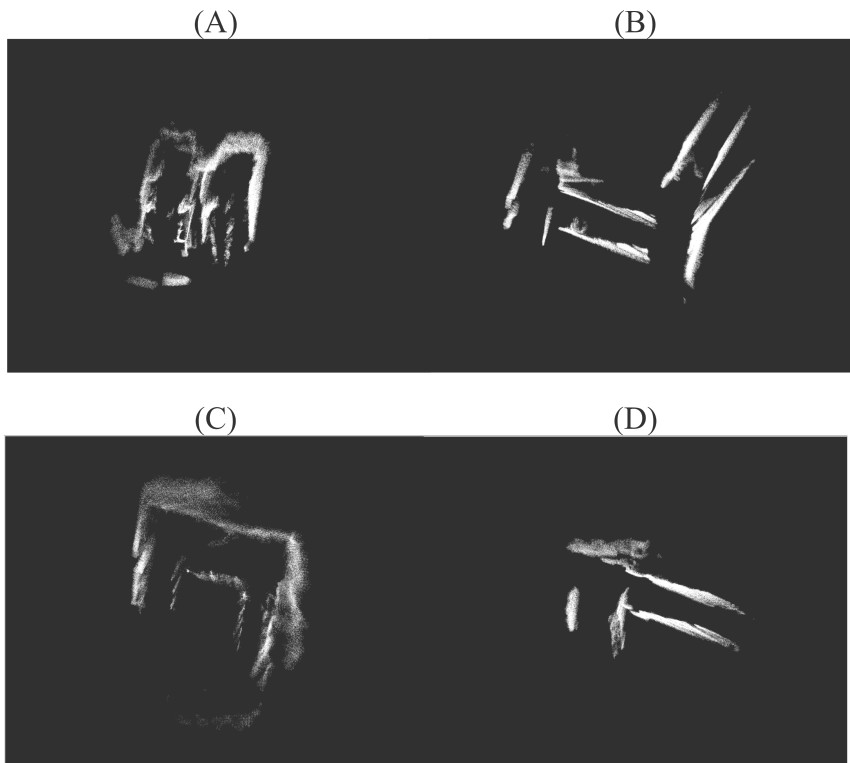

**Figure 9.** Example of the dense point cloud map for (**A**) the U-shaped hallway; (**B**) the H-shaped hallway; (**C**) the Loop-shaped hallway; (**D**) the T-shaped hallway.

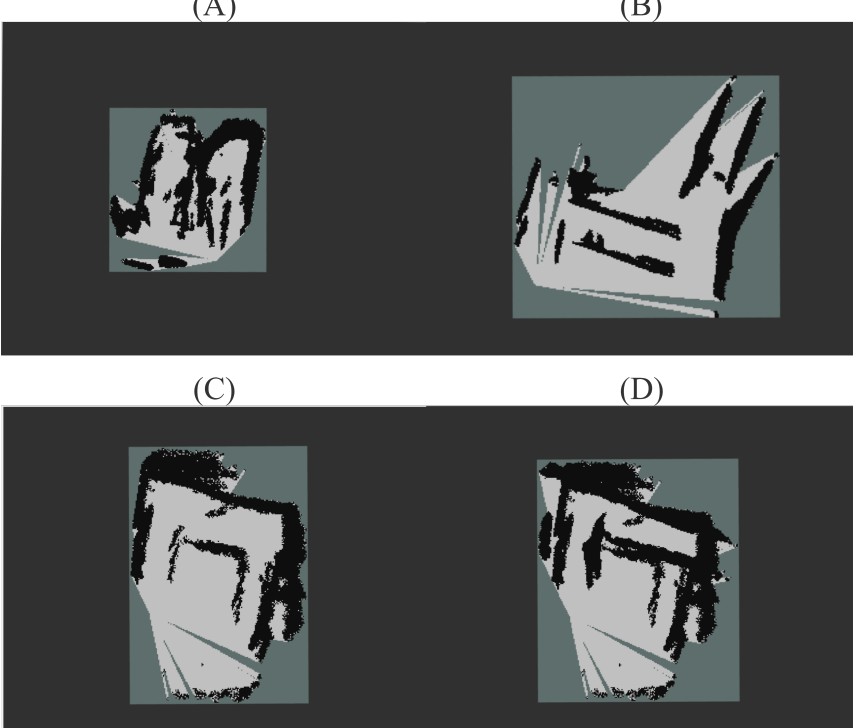

**Figure 10.** Example of the generated occupancy grid map of (**A**) the U-shaped hallway; (**B**) the H-shaped hallway; (**C**) the Loop-shaped hallway; (**D**) the T-shaped hallway.

#### 4.2.5. Experimental Data Analysis

For the above four scenes, the throughput of the map buildings is listed in Table 2. The number of key frames per minute can reflect the sensitivity of system recognition; our metric, matches per minute, indicates the average number of critical points recognized per minute. The table shows that the system exhibits a high number of keyframes per performance. The system can identify an average of about 86 effective feature points per minute, which shows that the system can function in a timely and effective manner. In summary, the system achieved a real-time operation speed in the map-building experiments.

**Table 2.** The map building throughput of the proposed system.

| Name | Speed Specification | |
| :---: | :---: | :---: |
| | Keyframes Number | Matches |
| A | 117 | 195 |
| B | 85 | 408 |
| C | 46 | 251 |
| D | 100 | 176 |

#### 4.3. Navigation

In each scenario, we define a specific shaped route traversed by the experimenter wearing a blindfold and holding our proposed cane. During the person's travel, the RGB-D camera receives color image information and depth information in real-time, while the global planner turns on the SLAM map building module to construct the map. The local navigation effects corresponding to each of the experiments are given below.

Figure 11 shows that the navigation node starts initialization after acquiring the occupied grid ground. During initialization, the navigation node subscribes to the current orientation of the camera and the target orientation selected by the user, which is published by the map building node in real-time. The green arrow in the figure represents the current orientation of the user, and the blue arrow is the direction that prompts the user to move forward.

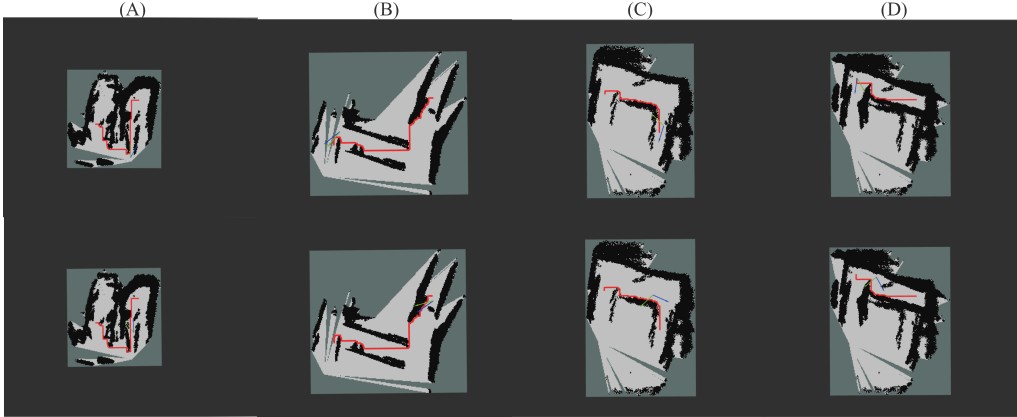

**Figure 11.** An illustration of the path planning and navigation in a (**A**) the U-shaped hallway; (**B**) the H-shaped hallway; (**C**) the Loop-shaped hallway; (**D**) the T-shaped hallway.

As shown in Scene (B), we defined a "Z" shaped path, with the obstacles on both sides of the corridor as obstacles. At the beginning of the experiment, we turn on the cane and send information about the starting point to the system, while the depth camera starts to record the path information and then stops recording when the path ends. The camera keeps collecting environmental information while the user walks with the cane. This recorded and documented path information can be used to complete the scene rebuilding

and navigation functions under the ORB-SLAM, as shown in the second set of Scene (B) images. Then, we use the ROS Navigation module for path navigation after building the map. We can find similar results in Scene (D), where a "U" shaped path is constructed.

For Scenes (A) and (C), we chose the laundry room and the classroom, which had a large number of obstacles. Before starting, we defined a "U" shaped path as shown in the first set of Scene (A) pictures, with the desks and chairs in the classroom, part of the walls, the laundry room walls, and the washing machine as obstacles. The experimenter walks the U-shaped route with a blind stick to complete the scene information collection. The navigation experiments show that our proposed system is effective when operating in a complex environment.

*4.4. Object Detection*

In detecting the targets, we obtained the data of the leading performance indicators as shown in Table 3. The mAP denotes the mean average precision [36], i.e., the mean value of Average Precision (*AP*), which is the primary evaluation metric for target detection algorithms. The computation of $mAP$ can be given by:

$$mAP = \frac{\sum_{i=1}^{n} AP_i}{n}, \tag{2}$$

where $AP_i$ is the *i*th average precision under a specific Intersection over Union (IoU) level, and $n$ is the total number of IoU levels.

**Table 3.** The performance comparison of various object detection algorithms.

| Method | mAP | FPS |
|---|---|---|
| Fast R-CNN | 70.0 | 0.5 |
| SSD321 | 45.4 | 16 |
| YOLOv3-320 | 51.5 | 45 |
| Our work | 55.3 | 35 |

The FPS and the mAP metrics usually describe the target detection model. FPS indicates how many frames (how many images) can be detected by the target network per second. Compared with similar algorithms, we can find that it is not easy to maintain a synergistic relationship between mAP and FPS values. mAP and FPS values of SSD321 are relatively low; Fast R-CNN has high mAP values but common FPS values; YOLOv3-320 has a high FPS value, but at the same time, the mAP value is low.

Experimental data show that our work can achieve a relatively high mAP value while maintaining a 35 FPS performance. This is because Fast R-CNN is a two-stage target detection algorithm that transforms the detection problem into a classification problem for local images by explicit region suggestion. This algorithm significantly improves the accuracy of the original target detection algorithm and has a lower miss detection rate but is very slow and has a large memory footprint. SSD is a one-stage target detection algorithm that does not directly generate the region of interest. It views the target detection task as a regression task on the whole image, providing an improved detection speed compared to Fast R-CNN. However, the SSD321 algorithm may produce duplicate frames, and the reduced ability for small targets can be another issue in some cases. In contrast, YOLOv3-320 [19] applies a single neural network to the whole image, which divides the image into different regions and thus predicts each region's bounding boxes and probabilities, which are weighted by the predicted probabilities. The algorithm is balanced in speed and accuracy and handles small object detection with high accuracy.

Our proposed target recognition algorithm employs a similar tactic to YOLO, which utilizes the global information of the image. Compared to YOLOv3-320, our model increases the amount of information input to the neural network by increasing the image

resolution and can slightly increase the model's accuracy. Hence, we conclude that the target monitoring function of this system meets the requirements of daily use, and can help users to recognize objects in a real-time manner.

## 5. Work Limitations

Our system has limitations in some respects and needs to be improved. For the proposed ORB-SLAM-based dense map reconstruction, deviation and mismatches can be introduced in the dense maps we build; hence the walking area needs a specific width to build a correct dense map. We consider developing an improved experimental tuning to enhance the system's graph-building performance. As for the path planning and navigation, the time complexity of our proposed PPA algorithm is $O(n^2 \cdot \log n)$. We may also consider improving the algorithm structure or employing some heuristic function to reduce the computational complexity. We currently use a model trained based on a generic dataset for target detection, which has limitations in its recognition capability in daily scenarios. We may consider redesigning the datasets to produce a better model with improved recognition accuracy.

## 6. Conclusions

This paper proposes a multi-sensory guidance system for the visually impaired. Our system can provide advice both in tactile and auditory sensations by using ORB-SLAM and YOLO techniques. Based on an RGB-D camera, the local obstacle avoidance system is realized at the tactile level through point cloud filtering that can inform the user via a vibrating motor. Our proposed method can generate a dense navigation map to implement global obstacle avoidance and path planning for the user through the coordinate transformation. Real-time target detection and a prompt voice system based on YOLO are also incorporated at the auditory level. The proposed system has been developed as a prototype cane. Experiments are performed using four different test scenarios. Experimental results show that the position and category of obstacles in the surrounding environment can be detected accurately in real-time. By integrating YOLO and ORB-SLAM, our proposed system can operate as a competent auxiliary to help visually impaired people to walk around securely.

**Author Contributions:** Conceptualization, Z.X. and Z.L.; data curation, Z.L.; formal analysis, Z.X. and Z.L.; funding acquisition, Z.X.; investigation, Z.X., Z.L., Y.Z., J.Z., F.L. and W.C.; methodology, Z.L.; project administration, Z.X.; resources, Z.X.; software, Z.L.; supervision, Z.X.; validation, Z.X. and Z.L.; visualization, Y.Z., J.Z., F.L. and W.C.; writing—original draft, Z.X., Z.L., J.Z., Y.Z.; writing—review & editing, Z.X., Z.L., Y.Z., J.Z., F.L. and W.C. All authors have read and agreed to the published version of the manuscript.

**Funding:** This research was funded by The Belt and Road Special Foundation of the State Key Laboratory of Hydrology-Water Resources and Hydraulic Engineering under Grant 2021490811.

**Institutional Review Board Statement:** The study was conducted according to the guidelines of the Declaration of Helsinki, and approved by the Institutional Review Board 14046.

**Informed Consent Statement:** Informed consent was obtained from all subjects involved in the study.

**Data Availability Statement:** All data were presented in the main text.

**Conflicts of Interest:** The authors declare no conflict of interest.

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
