# Peer review of "A Multi-Sensory Guidance System for the Visually Impaired Using YOLO and ORB-SLAM"

_information, doi:10.3390/info13070343_

Round 1

Reviewer 1 Report

The paper proposes a multi-sensory guidance system for the visually impaired. Authors have combined  YOLO and 12 ORB-SLAM to enable visually impaired persons safely (this is an important societal work).

The paper is well written, is clear to understand and is, to some point, well structured. Still, when a now model or approach is presented, in my opinion, its very important to have a "work limitations" section. Also it was important an more in-deep review of the art, before to present the proposed system.

The presented system is very interesting and with merit indeed. Its a fine work. But could (and should in my opinion) be improved with the mentioned issues. 

Reviewer 2 Report

In this paper, a multi-sensory guidance system for the visually impaired using YOLO and ORB-SLAM is proposed. This paper combines some algorithms together to accomplish a navigation and path planning task. This paper can be accepted after minor revision.

(1) The algorithm structure of ORB-SLAM and the brief introduction of YOLO are given. However, the algorithm flow of the whole process is lack, which is more interested to readers;

(2) The detail parameters of the experimental setups should be given clearly in a Table;

(3) What's the meaning of Fig.2. At least the axis units should be added.

(4) In Tab.2, what's the unit of mAP?

(5) In comparison part, only the comparison results are given, but the analysis should be added. Why the proposed method get the best result? What's the disadvantage of other tranditional methods?

(6)  It's better to analyze the computation complexity of the proposed method.
